# Rift Valley fever virus 78kDa envelope protein attenuates virus replication in macrophage-derived cell lines and viral virulence in mice

Kaori Terasaki[1,2]*, Birte Kalveram[3], Kendra N. Johnson[1], Terry Juelich[3], Jennifer K. Smith[3], Lihong Zhang[3], Alexander N. Freiberg[2,3,4,5,6], Shinji Makino[1,2,4,5,6]*

**1** Department of Microbiology and Immunology, The University of Texas Medical Branch, Galveston, Texas, United States of America, **2** Institute of Human Infection and Immunity, The University of Texas Medical Branch, Galveston, Texas, United States of America, **3** Department of Pathology, The University of Texas Medical Branch, Galveston, Texas, United States of America, **4** Center for Biodefense and Emerging Infectious Diseases, the University of Texas Medical Branch, Galveston, Texas, United States of America, **5** Center for Tropical Diseases, The University of Texas Medical Branch, Galveston, Texas, United States of America, **6** Sealy Institute for Vaccine Sciences, The University of Texas Medical Branch, Galveston, Texas, United States of America

* katerasa@utmb.edu (KT); shmakino@utmb.edu (SM)

**Data Availability Statement:** All relevant data are within the manuscript.

## Abstract

Rift Valley fever virus (RVFV) is a mosquito-borne bunyavirus with a wide host range including ruminants and humans. RVFV outbreaks have had devastating effects on public health and the livestock industry in African countries. However, there is no approved RVFV vaccine for human use in non-endemic countries and no FDA-approved antiviral drug for RVFV treatment. The RVFV 78kDa protein (P78), which is a membrane glycoprotein, plays a role in virus dissemination in the mosquito host, but its biological role in mammalian hosts remains unknown. We generated an attenuated RVFV MP-12 strain-derived P78-High virus and a virulent ZH501 strain-derived ZH501-P78-High virus, both of which expressed a higher level of P78 and carried higher levels of P78 in the virion compared to their parental viruses. We also generated another MP-12-derived mutant virus (P78-KO virus) that does not express P78. MP-12 and P78-KO virus replicated to similar levels in fibroblast cell lines and Huh7 cells, while P78-High virus replicated better than MP-12 in Vero E6 cells, fibroblast cell lines, and Huh7 cells. Notably, P78-High virus and P78-KO virus replicated less efficiently and more efficiently, respectively, than MP-12 in macrophage cell lines. ZH501-P78-High virus also replicated poorly in macrophage cell lines. Our data further suggest that inefficient binding of P78-High virus to the cells led to inefficient virus internalization, low virus infectivity and reduced virus replication in a macrophage cell line. P78-High virus and P78-KO virus showed lower and higher virulence than MP-12, respectively, in young mice. ZH501-P78-High virus also exhibited lower virulence than ZH501 in mice. These data suggest that high levels of P78 expression attenuate RVFV virulence by preventing efficient virus replication in macrophages. Genetic alteration leading to increased P78 expression may serve as a novel strategy for the attenuation of RVFV virulence and generation of safe RVFV vaccines.

**Funding:** This study was supported by Public Health Service grants AI127984 and AI148763 from the National Institutes of Health and Pilot grants from the Institutes for Human Infections and Immunology at The University of Texas Medical Branch. The funders had no role in study design, data collection and analysis, decision to publish, or preparation of the manuscript.

**Competing interests:** The authors have declared that no competing interests exist.

## Author summary

Rift Valley fever virus (RVFV) is the causative agent of Rift Valley fever, which is primarily endemic in African countries. The virus infects a wide variety of mammalian species, including ruminants and humans, and causes devastating effects on public health and the livestock industry. Because RVFV is transmitted by several mosquito species that are ubiquitous in non-endemic areas, there is a potential risk that the virus will spread outside of the current endemic areas. However, there are no effective therapeutics nor commercially available vaccines for human use. RVFV encodes a 78kDa protein (P78) of unidentified biological function in mammalian cells. We found that a mutant RVFV expressing a higher level of P78 (P78-High virus) showed lower infectivity and less efficient replication than parental RVFV in macrophage cell lines. P78-High virus and a mutant RVFV lacking P78 expression showed lower and higher virulence than the parental virus, respectively, in mice, suggesting that high levels of P78 expression prevent efficient virus replication in macrophages, leading to attenuation of virus virulence. Genetic alteration leading to increased P78 expression may serve as a novel strategy for the generation of attenuated RVFV as vaccine candidates.

## Introduction

Rift Valley fever virus (RVFV) is the causative agent for Rift Valley fever, a major public health concern in African countries. Since the virus was first identified in 1931 in Kenya [1], it has repeatedly caused outbreaks in African countries and outside the African continent (reviewed in [2]). RVFV also has a significant impact on the livestock industry because of its high mortality rate in young ruminants and high abortion rate in pregnant animals. Human RVFV infections generally manifest as self-limiting and nonfatal illnesses. However, a small percentage of patients develop encephalitis and hemorrhagic fever with a high mortality rate and suffer long-term neurological disease. Ocular diseases, which can result in permanent vision loss, are reported in up to 10% of infected humans [3], and association of the virus infection with miscarriage in pregnant women has been recently reported [4,5]. The virus has the potential to spread to almost any area of the world, including North America, via naturally occurring mosquito populations as RVFV can be transmitted by several different ubiquitous mosquito species [6,7]. RVFV is a Category A pathogen, which is the highest risk group to national security and public health according to the CDC/NIAID categorization of biodefense priority pathogens, and it is included in the WHO R&D blueprint list [8]. Currently, there is no approved RVFV vaccine for human use in non-endemic countries and no FDA-approved antiviral drug for RVFV treatment.

RVFV belongs to the family *Phenuiviridae*, genus *Phlebovirus*, and has a genome composed of three single-stranded, negative-sense or ambisense RNA segments, L, M, and S. The anti-genomic-sense L RNA encodes the RNA-dependent RNA polymerase (L protein). The anti-genomic-sense M RNA carries 5 in-frame AUGs within the 5' region and expresses two accessory proteins, 78-kDa protein (P78) and NSm from the 1st and 2nd/3rd AUG, respectively, and two major envelope glycoproteins, Gn and Gc from the 4/5th AUG, via a leaky scanning mechanism (Fig 1A). Gn and Gc are type I transmembrane glycoproteins and are expressed as a polyprotein precursor and cleaved by a signal peptidase. Gn and Gc interact with themselves and each other to form homodimers [9,10] and also form Gn-Gc heterodimers [11,12]. RVFV-like particles can be produced through the expression of Gn and Gc [13,14]. The S

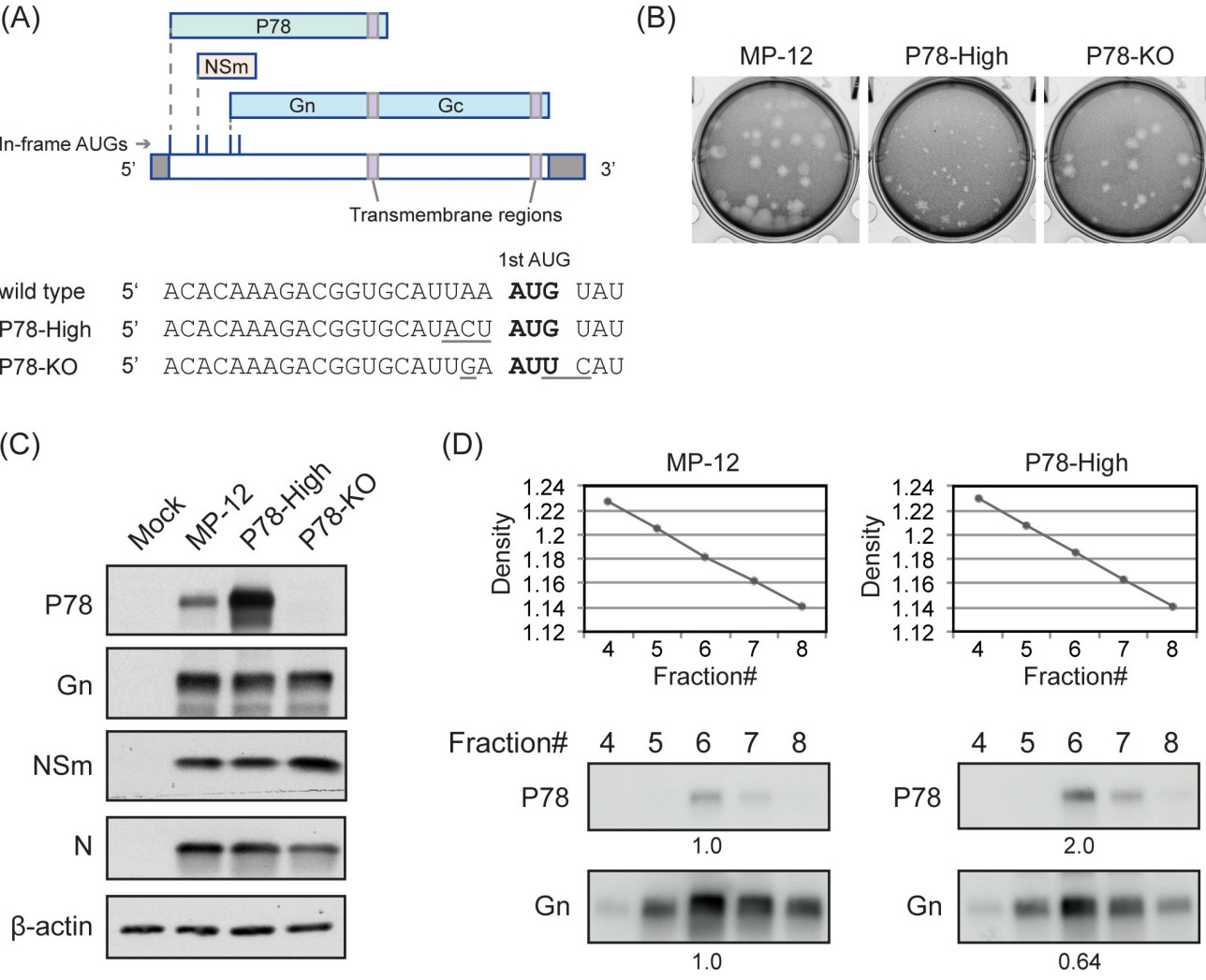

**Fig 1. Characterization of P78-High virus.** (A) Coding strategies of RVFV M segment and nucleotide sequence of mutated sites. RVFV M segment RNA is depicted in anti-genomic sense orientation. Underline indicates mutated sites. (B) Plaque morphologies of mutant viruses in Vero E6 cells. (C) Viral protein expression profiles. Vero E6 cells were infected with indicated viruses at MOI of 3 and harvested at 8 h p.i. Intracellular protein accumulation was analyzed by Western blot. (D) Western blot analysis of purified virus particles. Vero E6 cells were infected with the indicated viruses at MOI of 1. Culture supernatant was harvested at 16 h p.i. and subjected to sucrose gradient ultracentrifugation. Fractions were collected and sucrose densities were measured (top panels). Each fraction was subjected to Western blot for detection of P78 and Gn proteins using anti-NSm [15] and anti-Gn monoclonal (4D4) antibodies, respectively. MP-12 and P78-High samples were loaded on the same gel. Numbers indicate band intensities of P78 and Gn in P78-High samples relative to that in MP-12 samples.

segment uses an ambisense strategy for the expression of nucleocapsid (N) protein and an accessory protein, NSs.

P78 is a membrane glycoprotein [16,17] that shares the same amino acid sequence with Gn in the C-terminal region, while also having an amino acid sequence corresponding to pre-Gn region, between the 1st AUG and 4th AUG, at the N-terminal region (Fig 1A). P78 was incorporated into the virions of a virulent ZH501 strain replicated in mosquito C6/36 cells, whereas ZH501 virions that were released from Vero E6 cells carried an undetectable level of P78 [17]. Expression levels of P78 in RVFV-infected mosquito cells were higher than in mammalian cells [17], possibly because the flanking sequence of the 1$^{st}$ AUG, from which P78 is translated, is suitable for efficient translation in insect cells [18]. In contrast, the AUGs for NSm and Gn/

Gc expression are in an optimal Kozak context for efficient translation in mammalian cells [18]. Although P78 is not essential for virus replication in cell culture [15], P78 expression may be beneficial for virus replication in mammalian cells. Passaging an RVFV mutant, which does not express P78 via the removal of the 1st AUG in the M segment, in Vero E6 cells resulted in the emergence of a virus with a new in-frame AUG for P78 expression [18]. P78 is required for virus dissemination in the mosquito [18], whereas the biological functions of P78 for RVFV replication in mammals remains largely unknown. Another unknown is whether P78 affects pathogenesis of RVFV.

By using RVFV mutants expressing higher levels of P78 and another mutant lacking P78 expression, we explored the biological roles of P78 on virus replication in various mammalian cells and viral virulence in mice. We found that high levels of P78 expression interfered with efficient virus replication in macrophage cell lines and caused attenuation of virus virulence.

## Materials and methods

### Ethics statement

All mouse studies were performed in facilities accredited by the Association for Assessment and Accreditation of Laboratory Animal Care in accordance with the recommendations in the Guide for the Care and Use of Laboratory Animals (Institute of Laboratory Animal Resources, National Research Council, National Academy of Sciences, 1996). The animal protocols (protocol numbers, 1105023C and 2106038) were approved by the Institutional Animal Care and Use Committee of The University of Texas Medical Branch. All work with the wild type RVFV ZH501 and its variant were conducted in the Robert E. Shope and Galveston National Laboratory BSL-4 laboratories at The University of Texas Medical Branch (UTMB) in accordance with NIH guidelines and U.S. federal law.

### Cells

BSRT7/5 cells, which stably express T7 RNA polymerase [19], were maintained in Glasgow's minimal essential medium (MEM) (Gibco) containing 10% fetal bovine serum (FBS), 10% Tryptose Phosphate Broth, MEM Amino Acids Solution and geneticin (1 mg/ml). Vero E6 cells were maintained in Dulbecco's modified MEM (DMEM) (Gibco) containing 5% FBS. MRC-5 cells were maintained in Eagle's MEM (EMEM) (Gibco) containing 10% FBS, MEM Non-Essential Amino Acids Solution (Gibco), and 1% sodium pyruvate (Sigma). Huh7 cells, mouse embryonic fibroblasts (MEF), and NIH/3T3 cells were maintained in DMEM containing 10% FBS. Raw264.7 cells were maintained in DMEM containing 10% FBS and 1% sodium pyruvate. THP-1 cells (human monocyte-derived cells) were maintained in RPMI (Gibco) containing 10% FBS. Prior to virus infection, THP-1 cells were treated with 150 nM phorbol 12-myristate 13-acetate (PMA) for 48 h to promote differentiation to macrophage cells and rested in complete culture medium for 24 h. C6/36 cells were maintained in L-15 medium containing 10% FBS, 10% Tryptose Phosphate Broth, and L-glutamine at 28°C without $CO_2$. All media listed above was supplemented with 1 x penicillin/streptomycin (100 unit/mL of penicillin, 100 μg/mL of streptomycin) except for DMEM used for ZH501 experiments.

### Generation of recombinant RVFVs

A recombinant MP-12 strain, an attenuated RVFV strain obtained after 12 serial passages of RVFV ZH548 strain in the presence of 5-fluorouracil [20], and other MP-12-derived mutants were rescued by reverse genetics [21], except that BSRT7/5 cells were used in place of BHK/T7-9. Culture supernatant was collected from transfected BSRT7/5 cells and saved as P0 virus.

Rescued virus was further amplified by inoculating P0 virus into Vero E6 cells and collected as P1 virus. For sequencing analysis, total RNA was extracted from P1 virus and subjected to reverse transcription with random hexamers. The M segment sequence of rescued viruses were confirmed by Sanger sequencing using primers designed for the M segment. Titers of the rescued viruses were determined by standard plaque assay [21] in Vero E6 cells. P1 virus was used for all experiments in this study. To generate P78-High viruses, pProT7-M [21], which encodes anti-genomic sense M segment, was modified by a standard recombinant PCR method to generate pProT7-M carrying mutations in the 1st AUG flanking region. pProT7-M-P78-ACC, which carries the nucleotide sequence ACC in place of sequence TAA positioned at 18–20 in M segment, were generated and used for virus rescue. Rescued virus using the pProT7-M-P78-ACC was subjected to standard plaque assay using Vero E6 cells. Six clones were isolated from single plaques after three rounds of plaque cloning. Total RNA was extracted from Vero E6 cells infected with the plaque cloned viruses and subjected to reverse transcription with random hexamers, followed by PCR with a primer set (Forward: 5'-acacaaa-gacggtgc-3', Reverse: 5'-catcaaaactgctacatgcc-3') for amplification of the mutated region in M segment for DNA sequencing. For generation of P78-High virus, pProT7-M-P78-ACT plasmid, which carries nucleotide sequence ACT in place of sequence TAA positioned at 18–20 in M segment, was generated and used for virus rescue. To generate a mutant virus lacking the 1st AUG in the M segment (P78-KO virus), which was named as arMP-12-del78 in our previous study [15], mutated pProT7-M plasmid, which carries nucleotide sequence GAAUUC positioned at 19–24 in M segment [15], was used for virus rescue.

A recombinant RVFV ZH501 strain and its variant were generated in the same way as MP-12 based viruses, except that the reverse genetics system for ZH501 [22] was used for virus rescue and all procedures were performed in a BSL-4 laboratory at UTMB.

## Western blot analysis

Cells were suspended in SDS polyacrylamide gel electrophoresis (SDS-PAGE) sample buffer after being washed with phosphate-buffered saline (PBS). Samples were boiled for 5 min (MP-12 samples) or 15 min (ZH501 samples) and subjected to SDS-PAGE. Proteins were electro-blotted onto polyvinylidene difluoride membranes (immune blot: Bio Rad). After blocking with skim milk for 1 h at room temperature, the membranes were incubated with the primary antibody overnight at 4˚C, followed by incubation with the secondary antibody for 1 h at room temperature. The membrane was washed three times with PBS containing 0.01% Tween 20 between each incubation step. ECL Western Blotting Detection Reagent (GE Healthcare Life Sciences) or ECL plus Western Blotting Substrate (Pierce) was used for detection of blotted proteins. For detection of Gn, NSm, and N proteins, anti-Gn mouse monoclonal (4D4) [23], anti-NSm [15] and anti-N [15] antibodies were used. Anti-NSm antibody was also used for detection of P78, as it detects both P78 and NSm [15].

## Flow cytometry

Virus-infected cells were detached from the dish by Accumax (Innovative Cell Technologies) at 8 h post infection (p.i.) and suspended in culture media. After washing with PBS containing 1% bovine serum albumin (BSA), cells were fixed with 2% formaldehyde/PBS for 30 min at room temperature and blocked with blocking buffer (PBS containing 0.2% saponin and 1% BSA for Raw264.7 cells, Vero E6 cells, and Huh-7 cells; PBS containing 0.1% saponin and 1% BSA for THP-1 and C6/36 cells) for 15 min on ice. Cells were incubated with mouse monoclonal antibody 4D4 (anti-Gn antibody) for 30 min on ice in the presence of 0.5% saponin and 1% BSA. After washing with the blocking buffer, cells were labeled by Alexa fluor

488-conjugated anti-mouse antibody for 30 min in the presence of 0.5% saponin and 1% BSA for Raw264.7, Vero E6, and Huh-7 cells or 0.1% saponin and 1% BSA for THP-1 and C6/36 cells. The cells were washed with PBS containing 1% BSA, passed through a cell strainer (BD Falcon), and analyzed on an LSRII Fortessa (BD Biosciences). Single cells were gated based on their forward scatter and side scatter profile. More than 30,000 counts of the gated single cells were analyzed for each experiment.

## Virus binding and virus internalization assays

Three stocks of each virus were prepared separately and used for three independent binding assays. Prior to virus infection, the virus samples were separated into two groups. One was used for virus inoculation and the other was suspended in TRIzol LS and used for measuring amounts of viral L RNA in the inoculum. For virus binding assay, cells were inoculated with each virus at a multiplicity of infection (MOI) of 3 and incubated for 1 h on ice. Cells were then were washed with Hank's Balanced Salt Solution (HBSS) three times and suspended in TRIzol reagent. For virus internalization assay, cells that had been incubated with the virus for 1 h at 37˚C, or 28˚C for C6/36 cells, were washed with HBSS three times and then incubated with 0.5 mg/ml of pronase (Sigma Aldrich) for 10 min at room temperature to remove extracellular viruses. After washing with HBSS three times, cells were suspended in TRIzol reagent. Total RNA was extracted from the harvested samples according to the manufacturer's instructions and subjected to RT-qPCR as described below. Copy numbers of L RNA obtained from virus inoculated cells were normalized by copy numbers of L RNA in the input samples.

## RT-qPCR

RT-qPCR was performed for the detection of genomic sense L RNA as described previously [24]. To generate *in vitro*-synthesized genomic sense L RNA for copy number calculation, pProT7-L(-) encoding genomic sense L segment [21] was linearized by NotI digestion and subjected to RNA synthesis using MEGAscript T7 transcription kit (Applied Biosystems/Invitrogen). A known concentration of *in vitro*-synthesized L RNA was serially diluted and subjected to cDNA synthesis using Superscript-III first strand synthesis system (Invitrogen) with an RT primer, which binds to genomic sense L RNA and carrying the non-viral sequence (tagged RT primer: 5'-GGCCGTCATGGTGGCGAATAgcccaatcatggattctat-3'). For experimental samples, 400 ng of total RNA was subjected to cDNA synthesis using the tagged RT primer and purified by the PCR purification kit (Qiagen). qPCR was performed with a PCR primer set for detection of genomic sense L RNA (Forward primer: 5'-GGCCGTCATGGT GGCGAATA-3', reverse primer: 5'-ccagattgaagtctatgg-3') using SsoAdvanced Universal SYBR Green Supermix (Bio-rad). Copy numbers of L RNA in experimental samples were calculated according to the copy number of *in vitro* synthesized L RNA.

## Purification of virus particles

Culture supernatants harvested from cells infected with ZH501-derived viruses, but not infected with MP-12-derived viruses, were irradiated with 5 MRad of $^{60}$Co gamma radiation to inactivate viruses prior to virus purification. Harvested culture supernatant from infected Vero E6 cells was clarified by centrifugation at 3,000 rpm for 15 min. Sucrose solution were prepared in NTE buffer consisting of 0.1 M NaCl, 10 mM Tris-HCl, pH 7.2, and 1 mM EDTA. The clarified supernatant was layered on top of a step sucrose gradient consisting of 20, 30, 50, and 60% sucrose (wt/vol) and centrifuged for 3 h at 26,000 rpm at 4˚C using a Beckman SW28 rotor. The virus particles at the interface of 30 and 50% sucrose were collected and diluted with NTE buffer. Samples were layered on top of a continuous sucrose gradient consisting of

20–60% sucrose and centrifuged for 18 h at 4°C. Twelve fractions were collected from the bottom of the tube and pelleted down through a 20% sucrose cushion at 38,000 rpm for 2 h at 4°C using a Beckman SW41 rotor.

### Testing virulence of MP-12-derived mutant viruses in young mice

Pregnant CD-1 mice were purchased from Charles River Laboratories. Sixteen-day-old mice were intraperitoneally inoculated with $10^4$ PFU of MP-12, P78-High, or P78-KO virus (or with HBSS) and were observed for survival for 21 days post inoculation.

### Testing virulence of ZH501 and ZH501-P78-High virus in adult mice

Five-week-old female CD1 mice were purchased from Charles River Laboratories. Mice were intraperitoneally inoculated with 10 or $10^2$ PFU of recombinant ZH501, ZH501-P78-High virus, or with PBS (n = 10 for each group). Clinical signs of disease and body weight were monitored throughout the 21 days duration of the study. The grade of clinical disease was scored as follows: 1- healthy; 2- lethargic, ruffled fur; 3- score 2 + hunched posture, orbital tightening; 4- score 3 + reluctance to move when stimulated, paralysis, unable to access feed and water normally, moribund appearance or ≥20% weight loss. Mice that were assigned a score of 4 were immediately euthanized for humane reasons and were reported as dead the following day.

## Results

### Generation and characterization of P78-High virus

To obtain a mutant virus that expresses increased levels of P78 in mammalian cells, we initially aimed to generate an MP-12-derived mutant virus carrying "ACC" in place of "UAA" at nucleotide position 18–20, immediately upstream of the 1st AUG of the antigenomic M segment RNA (Fig 1A); replacing UAA with ACC ensures that the 1st AUG is in a Kozak context [25]. Using a plasmid expressing antigenomic-sense M segment carrying this mutation and other plasmids for a reverse genetics system [21], we rescued viruses forming plaques of various sizes in Vero E6 cells. Sequence analysis of six plaque-cloned isolates showed that three had "ACU" and two had "UAC", in place of the introduced "ACC", and only one retained the introduced "ACC", implying that the virus carrying the "ACC" mutation was genetically unstable and/or had poor replication fitness.

We next introduced the "ACT" sequence immediately upstream of the 1st AUG in the plasmid expressing M segment RNA (Fig 1A) and rescued a mutant virus. The rescued virus (referred to as P78-High virus), which was amplified once in Vero E6 cells, retained the introduced ACU sequence and formed smaller size plaques as compared with MP-12 in Vero E6 cells (Fig 1B). The P78-High virus retained the introduced "ACU" mutation after 5 serial passages in MRC-5 cells. We also rescued P78-KO virus, which lacked the 1st AUG in the M segment [15], using a reverse genetics system.

We examined intracellular accumulation of P78, Gn, N, and NSm, the latter of which is translated from the 2nd/3rd AUGs in the M mRNA, in Vero E6 cells infected with MP-12, P78-High virus, or P78-KO virus (Fig 1C). Replication of P78-High virus resulted in efficient accumulation of P78, the amount of which was higher than in MP-12-infected cells. Accumulation of Gn was slightly lower in the P78-High virus-infected cells than in MP-12-infected cells (Fig 1C), suggesting that usage of the 4th/5th AUGs for Gn/Gc expression in P78-High virus was less efficient than in MP-12, probably due to the introduction of an optimal translation context for the 1st AUG in the P78-High virus. As expected, P78 accumulation did not

occur in P78-KO virus-infected cells. All three viruses accumulated similar levels of NSm, which is translated from the 2nd/3rd AUG. Similar levels of N protein accumulation occurred in MP-12-infected cells and P78-High virus-infected cells, while P78-KO virus-infected cells accumulated slightly lower levels of N protein.

To test whether the P78 is incorporated into P78-High virus particles, we purified virus particles released from MP-12-infected Vero E6 cells and those released from P78-High virus-infected Vero E6 cells after infection at an MOI of 1 by discontinuous sucrose gradient centrifugation followed by continuous sucrose gradient centrifugation. Panel Fig 1D shows sucrose densities (top panels) and amounts of P78 and Gn in each fraction (bottom panels). P78 was co-sedimented with Gn in both MP-12 and P78-High virus, strongly suggesting that P78 was assembled into virus particles. After pelleting down purified MP-12 particles and P78-High virus, we suspended the samples in sample buffer, applied them to the same gel, and performed Western blot analysis to detect P78 and Gn protein in the purified viruses (Fig 1D, bottom panels). The amounts of P78 and Gn in the purified P78-High were 200% higher and 36% lower than those in the purified MP-12, respectively. When normalized by the amount of Gn, P78 signal was approximately 3 times higher in purified P78-High virus than in purified MP-12, suggesting that P78-High virus particles carried higher amounts of P78 than MP-12 particles.

Although a previous study reported that a virulent RVFV strain, ZH501, amplified in Vero E6 cells contained undetectable levels of P78 [17], we suspected that the ZH501 virion could incorporate P78 if intracellular P78 expression was increased. To test this possibility, we generated wt RVFV strain ZH501-derived mutant carrying the same "ACU" sequence immediately upstream of the 1$^{st}$ AUG in M segment (ZH501-P78-High virus). Accumulation of P78 in ZH501-P78-High virus-infected cells was higher than that in ZH501-infected cells (Fig 2A), while no changes in expression levels of other viral proteins were observed in the two virus samples. Co-sedimentation of P78 and Gn was observed in both ZH501 and ZH501-P78-High virus samples, implying assembly of P78 in both virus particles (Fig 2B). The amounts of P78 in purified virions of these two viruses roughly correlated with intracellular accumulation levels of P78 in infected cells.

## Effects of P78 on virus replication kinetics and virus infectivity

To determine whether the presence of high levels of P78 in the virion and/or accumulation of high levels of intracellular P78 affect virus replication, we inoculated MP-12, P78-High virus or P78-KO virus at an MOI of 0.01 into various cell lines and examined replication kinetics (Fig 3A). Although plaques that were formed by P78-High virus were smaller in size than those produced by MP-12 in Vero E6 cells (Fig 1B), P78-High virus replicated as efficiently as MP-12 in Vero E6 cells late in infection. P78-High virus showed higher titers than MP-12 up to 24 h p.i. in human lung fibroblast MRC-5 cells, after 24 h p.i. in MEF cells and NIH-3T3 cells, another murine embryonic fibroblast cell line, or throughout infection in Huh-7 cells, a hepatocyte-derived human carcinoma cell line. In marked contrast, P78-High virus replicated less efficiently than MP-12 in a murine macrophage cell line, Raw 264.7. To ascertain whether P78-High virus also replicates inefficiently in another macrophage cell line, we treated a human monocyte-derived cell line, THP-1 cells, with PMA to induce differentiation to macrophages [26], and infected PMA-treated THP-1 cells with these viruses. Due to inefficient virus replication in PMA-treated THP-1 cells after low MOI infection, we inoculated each virus in PMA-treated THP-1 cells at an MOI of 3. Productive replication of MP-12 occurred in PMA-treated THP-1 cells, whereas P78-High virus replicated very inefficiently. MP-12 showed higher titers than the P78-KO virus in Vero E6 cells throughout infection, up to 24 h p.i. in MRC-5 cells, and after 24 h p.i. in NIH-3T3 cells, whereas P78-KO virus replicated more

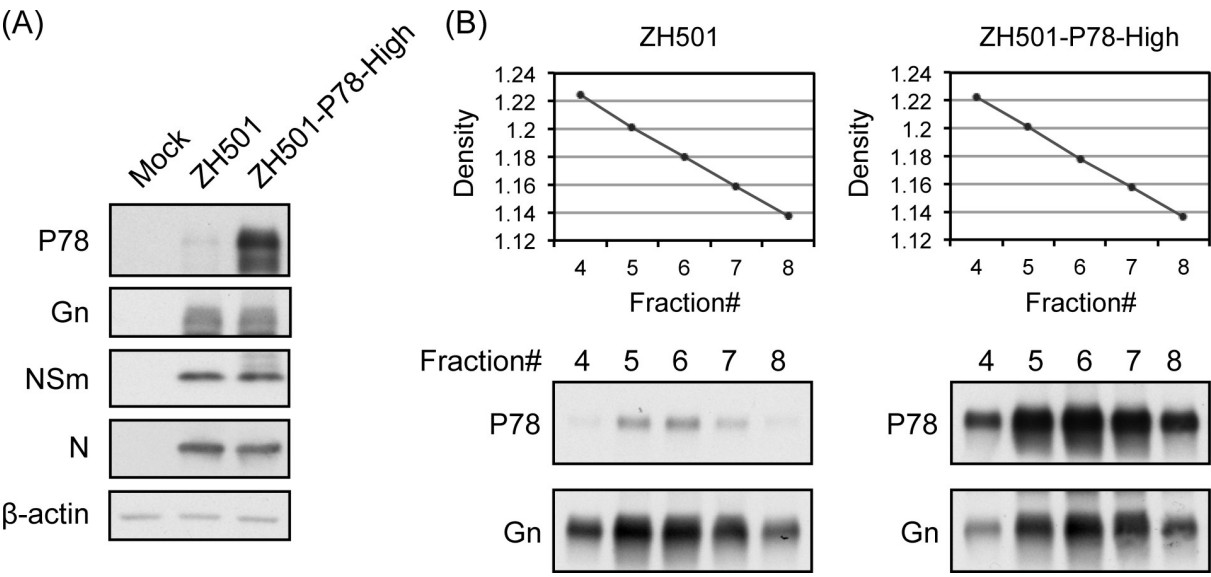

**Fig 2. Characterization of RVFV virulent strain ZH501 and its mutant ZH501-P78-High virus.** (A) Vero E6 cells were infected with the indicated viruses at an MOI of 5 and harvested at 24 h p.i. Intracellular protein accumulation was analyzed by Western blot. (B) Western blot analysis of purified virus particles. Vero E6 cells were infected with the indicated viruses at an MOI of 1. Culture supernatant was harvested at 24 h p.i. and subjected to sucrose gradient ultracentrifugation. Fractions were collected and sucrose densities were measured (top panels). Each fraction was subjected to Western blot for detection of viral proteins.

efficiently than MP-12 throughout the infection in Raw 264.7 and at late times p.i. in PMA-treated THP-1 cells. In C6/36 mosquito cells, P78-High and P78-KO viruses replicated less efficiently than MP-12. In summary, MP-12 and P78-KO virus replicated to similar levels in fibroblast cell lines and Huh 7 cells, and P78-High virus replicated better than MP-12 in non-macrophage cell lines. Notably, P78-High virus and P78-KO virus replicated less efficiently and more efficiently, respectively, than MP-12 in macrophage cell lines.

Analysis of replication kinetics of ZH501 and ZH501-P78-High virus in VeroE6 cells and macrophage cell lines (Fig 3B) showed that both replicated efficiently in VeroE6 cells, whereas ZH501-P78-High virus replicated less efficiently than ZH501 throughout infection in THP-1 cells and later in infection in Raw264.7 cells. These data showed that both P78-High virus and ZH501-P78-High virus replicated less efficiently than their parental viruses in macrophage cell lines.

Because P78 is exposed on the surface of RVFV particles [17], we tested whether altering the amount of P78 on the virion surface could affect virus infectivity by using MP-12, P78-High virus, or P78-KO virus to infect Vero E6, Raw 264.7, or Huh-7 cells at an MOI of 0.1 and counting the number of cells expressing Gn at 8 h p.i. Because Gn protein is encoded in M segment and both L and N proteins are needed for viral RNA replication/transcription [27–29], Gn expression only occurs in cells infected with virions carrying all three RNA segments. Accordingly, counting the number of Gn-expressing cells indicates the number of cells that were successfully infected with fully infectious virus. Inoculation of P78-High virus, MP-12, and P78-KO virus into Raw 264.7 cells resulted in 7.2%, 13.4%, and 21.6% of Gn-positive cells, respectively (Fig 4A). In contrast, Vero E6 cells independently inoculated with P78-High virus, MP-12, and P78-KO showed 24.0%, 19.5%, and 8.0% of Gn-positive cells, respectively (Fig 4B). These results demonstrated that viruses that underwent efficient virus replication exhibited a higher infectivity in Vero E6 cells and Raw 264.7 cells. Inoculation of P78-High virus and P78-KO virus into Huh7 cells resulted in a similar number (~15%) of Gn-positive cells,

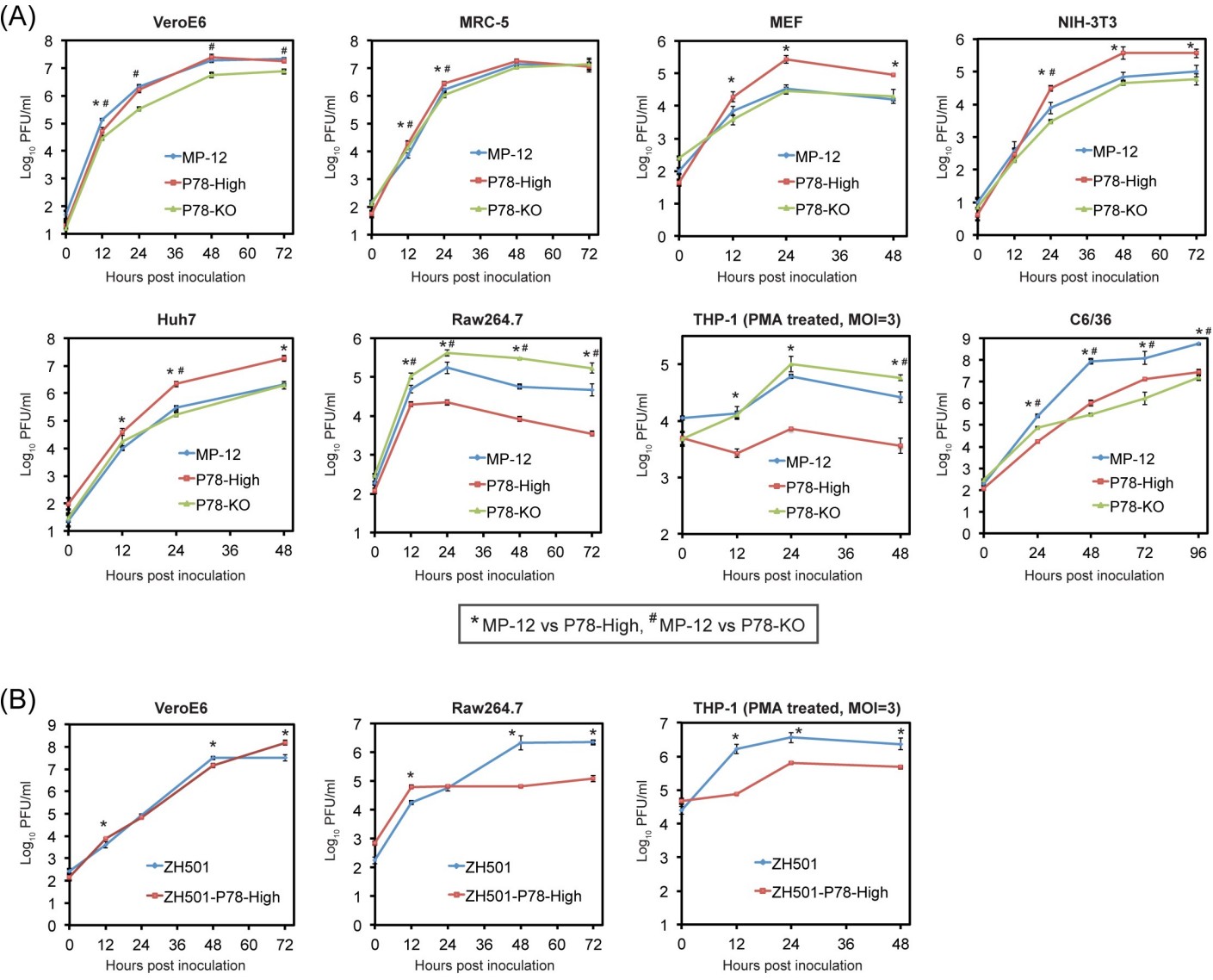

**Fig 3.** Growth kinetics of MP-12, P78-High virus, and P78-KO virus (A) and ZH501 and ZH501-P78-High (B) in different cell lines. Each of these viruses was inoculated into the indicated cell lines at an MOI of 0.01, except that viruses were inoculated at an MOI of 3 into THP-1 cells. Culture supernatant was collected at the indicated time points and infectivity determined by plaque assay in Vero E6 cells. The difference in virus titers between the parental virus and its variants was assessed by unpaired t-test at each time point. The data represents the mean values ± standard deviation from three independent experiments. Statistically significant differences (P<0.05) in virus titers between MP-12 and P78-High virus and between ZH501 and ZH501-P78-High virus are indicated by * and those in virus titers between MP-12 and P78-KO virus are indicated by #.

while MP-12 inoculation resulted in a higher percentage of Gn-positive cells at 19.2% (Fig 4C). Although P78-High virus replicated better than MP-12 in Huh7 cells, no correlation between virus replication kinetics and virus infectivity was observed in this cell line.

Infection of these viruses in C6/36 cells at an MOI of 0.1 and THP-1 cells at an MOI of 0.5 showed low infectivity, preventing us to confidently evaluate each virus' infectivity in these cells. Accordingly, we inoculated each virus at an MOI of 1.0 and 3.0 to C6/36 cells and THP-1 cells, respectively, to determine virus infectivity in these cells, although these experimental conditions might have caused co-infection of multiple virus particles in some cells. Because Gn was not detected by IFA at 8 h p.i. in C6/36 cells, likely due to delayed protein expression

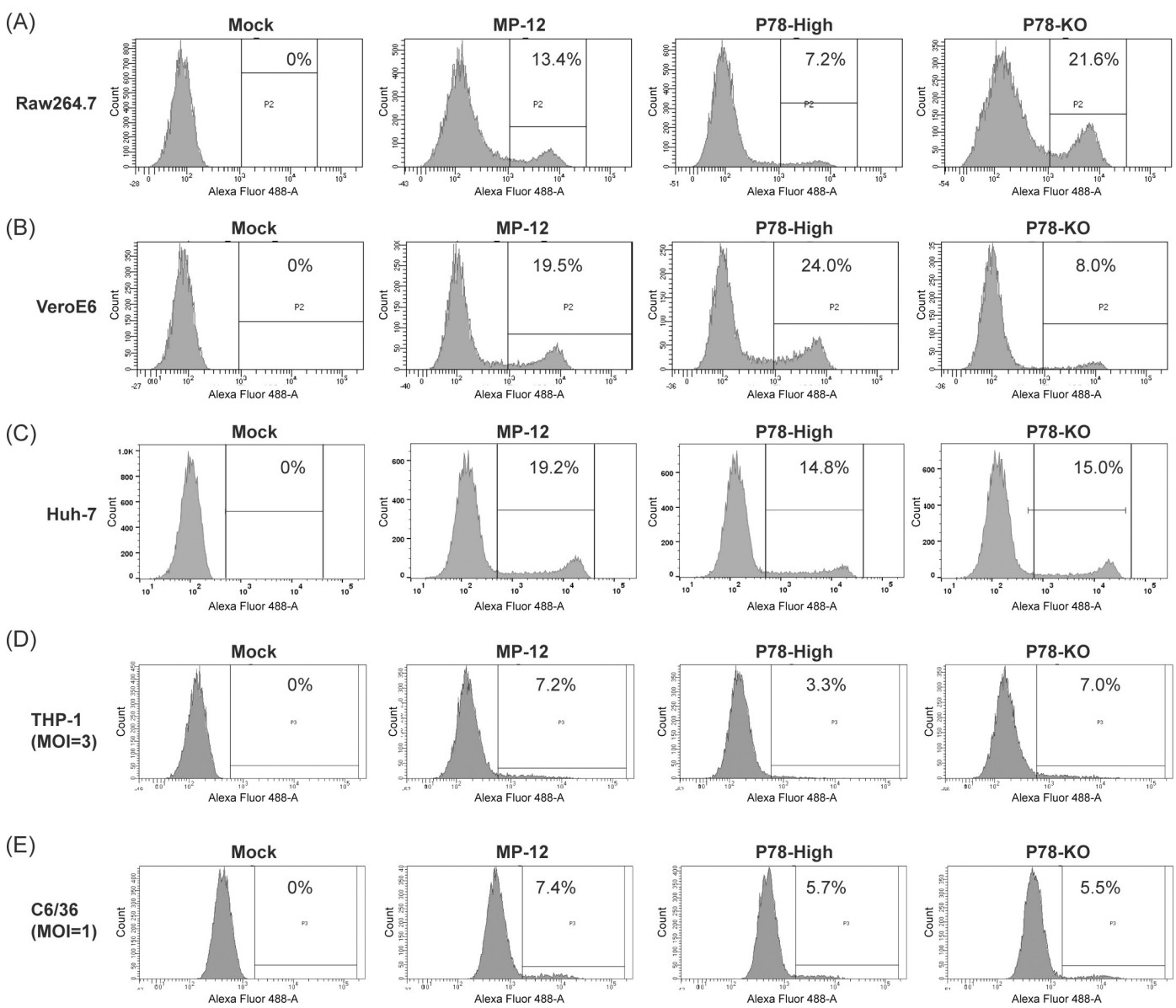

**Fig 4. Infectivity of MP-12 and its variants in different cell lines.** (A) Raw264.7, (B) Vero E6, and (C) Huh-7 cells were infected with the indicated viruses at an MOI of 0.1. (D) THP-1 cells and (E) C6/36 cells were infected with the indicated viruses at an MOI of 3 and 1, respectively. Cells were fixed at 8 h p.i. or 24 h p.i. for C6/36 cells and permeabilized. Cells were incubated with anti-Gn monoclonal antibody (4D4), followed by Alexa fluor 488-conjugated anti-mouse antibody, and analyzed by flow cytometry. The numbers shown in each panel represents the percentage of virus-infected cells.

at 28˚C, C6/36 cells were fixed at 24 h p.i. for flow cytometry analysis. Even using high MOI, these viruses showed low infectivity in C6/36 cells (Fig 4E), yet MP-12, which replicated more efficiently than P78-KO virus and P78-High virus in this cell line (Fig 3), showed higher infectivity than P78-KO virus and P78-High virus in C6/36 cells. All three viruses exhibited low infectivity in THP-1 cells (Fig 4D) and the differences in infectivity among them were marginal. Yet, like Raw 264.7 cells, infectivity was lowest for P78-High virus in THP-1 cells. These data showed that RVFV mutants expressing high levels of P78 in the virion and in cells replicated less efficiently and demonstrated lower infectivity than their parental viruses in macrophage cell lines.

### Effects of virion P78 on virus binding and virus internalization into cells

We next examined whether P78 affects early steps in virus replication, including virus binding and virus internalization into the cells, by using Vero E6, Raw 264.7, Huh7, and C6/36 cells (Fig 5). For the virus binding assay, we inoculated each virus at an MOI of 3 and incubated the cells for 1 h on ice to allow virus binding, but not virus internalization, to the cells. The levels of virus particles that attached to the cells were estimated by measuring copy number of viral L RNA in total cellular RNAs by RT-qPCR. Because each of these viruses may have different amounts of L RNA segment in the inoculum, we also measured the amounts of the L RNA segment in the inoculum and used them to normalize amounts of L RNA levels in the virus-inoculated cells. Virus internalization assays were performed by the same method used for virus

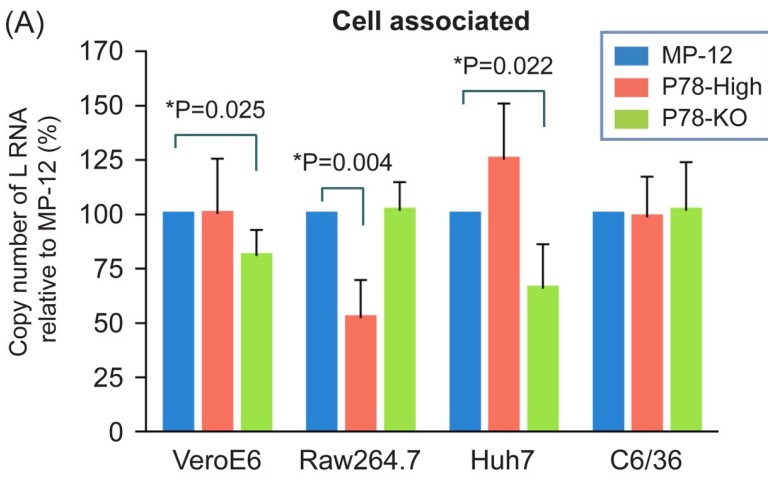

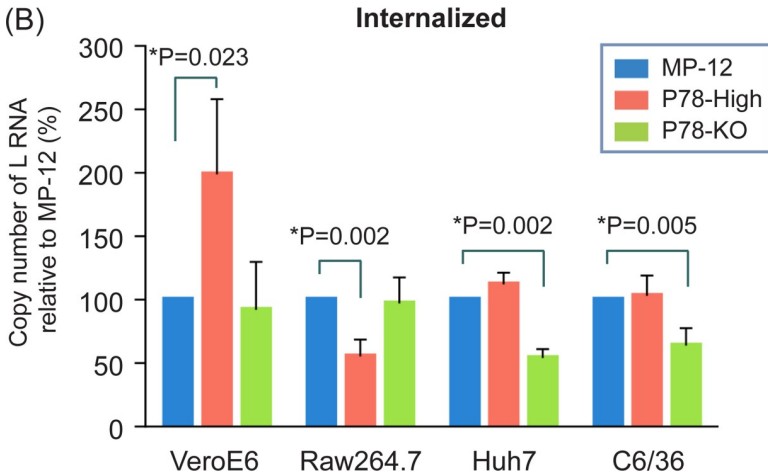

**Fig 5.** Efficiency of virus binding (A) and internalization (B) of MP-12 and its variants in Vero E6, Raw 264.7, Huh7, and C6/36 cells. (A) Cells were inoculated with the indicated viruses and incubated for 1 h at 4˚C on ice. After removal of unadsorbed viruses, total RNA was extracted from the cells and subjected to RT-qPCR for detection of genomic sense L RNA. Viral RNA was also extracted from inoculum virus as an input sample and subjected to RT-qPCR. Copy number of L RNA in experimental samples were normalized by copy number of L RNA in input samples. (B) Cells were inoculated with the indicated viruses and incubated for 1 h at 37˚C or 28˚C for C6/36 cells. Cells were further treated with pronase to remove cell-associated extracellular virus. Total RNA was extracted from the cells and subjected to RT-qPCR for detection of viral L RNA. Data was normalized to input samples as described in (A). The difference between MP-12 and its variants was assessed by unpaired t-test. The data represents the mean values ± standard deviation from three independent experiments.

binding assay, except that virus adsorption was performed at 37˚C for 1 h to allow virus internalization into the cells.

P78-High virus bound to Raw 264.7 cells less efficiently than MP-12 (Fig 5A). Likewise, virus internalization assays showed that amounts of intracellular L RNA of P78-High virus were also lower than those of intracellular L RNA of MP-12 (Fig 5B). Because P78-High virus replicated less efficiently and had a lower infectivity than MP-12 in Raw 264.7 cells (Figs 3 and 4), these data suggest that inefficient binding of P78-High virus to Raw 264.7 cells led to inefficient virus internalization and virus replication. P78-High virus and MP-12 bound to Vero E6 cells at a similar efficiency, whereas P78-High virus was internalized more efficiently than MP-12. These data imply that efficient internalization of P78-High virus contributed to the higher infectivity of P78-High virus compared with MP-12 in Vero E6 cells. We noted that P78-KO virus bound to Vero E6 cells less efficiently than MP-12, whereas both viruses had similar levels of virus internalization. The data imply that the absence of P78 in the virion had a negative impact on virus binding, but not virus internalization, to Vero E6 cells. Although MP-12 and P78-KO virus were internalized at similar levels in Vero E6 cells, the former replicated more efficiently (Fig 3) and had a higher infectivity than the latter in this cell line (Fig 4). These data suggest that an unidentified post virus internalization step(s) contributed to the efficient replication of MP-12 in Vero E6 cells. P78-KO virus bound and internalized less efficiently than MP-12 in Huh7 cells, which probably caused less efficient infectivity of P78-KO virus than MP-12 in Huh7 cells (Fig 4). Although P78-High virus replicated more efficiently than MP-12 in Huh7 cells (Fig 3), no significant difference in virus binding and virus internalization was detected between P78-High virus and MP-12. In C6/36 cells, P78-High virus and MP-12 showed similar binding efficiency, whereas P78-High virus showed lower internalization efficiency than MP-12, demonstrating that the absence of P78 negatively affected virus internalization in this cell line.

## P78-High and ZH501-P78-High viruses exhibit attenuated virulence in mice

To test whether P78 expression level affects virus virulence, we intraperitoneally inoculated HBSS or $10^4$ PFU of MP-12, P78-High virus, or P78-KO virus into 16-day-old CD-1 mice and observed the mice for survival for 21 days p.i. (Fig 6). Although none of the mice inoculated

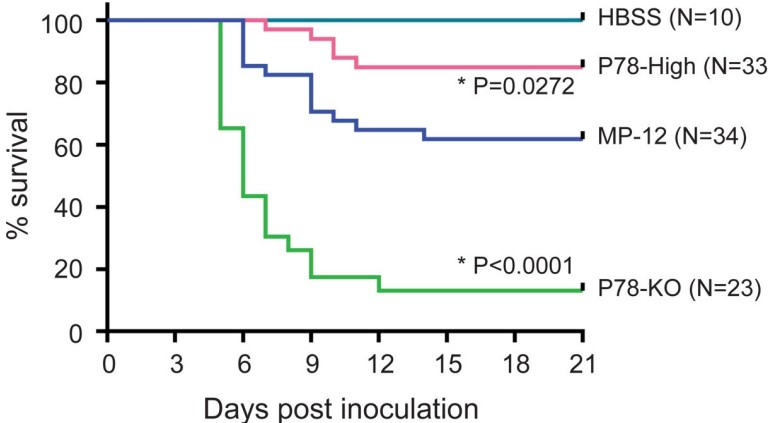

**Fig 6. Virulence of MP-12 mutants in young mice.** Sixteen-day-old CD-1 mice were inoculated with HBSS or $10^4$ PFU of the indicated viruses intraperitoneally and observed for 21 days after inoculation. Differences in survival curves between MP-12 and its variants were analyzed by log-rank test.

with HBSS died, inoculation of MP-12, P78-High virus, and P78-KO virus resulted in the death of 38%, 15%, and 87% of the mice, respectively; mortalities of P78-High virus-inoculated mice and P78-KO virus-inoculated mice were statistically lower and higher, respectively, than MP-12-inoculated mice (Fig 6). Also, the onset of death induced by MP-12 was later and earlier than that induced by P78-KO virus and P78-High virus, respectively (Fig 6). These data demonstrated that higher levels of P78 expression attenuated virus virulence, while the absence of P78 expression led to enhancement of virulence.

To determine whether ZH501-P78-High virus also shows lower virulence than its parental virus, ZH501, five-week-old CD-1 mice were intraperitoneally inoculated with 10 or 100 PFU of ZH501 or ZH501-P78-High virus. Mice in the control group were intraperitoneally inoculated with PBS. Inoculation of 10 or 100 PFU of ZH501 resulted in death of 90% of the mice whereas inoculation of 10 PFU and 100 PFU of ZH501-P78-High virus caused death of 40% and 70% of the mice, respectively (Fig 7A). The differences between the survival curves of the 10 PFU groups were statistically significant. In 10 PFU group, mice inoculated with ZH501-P78-High virus did not show pronounced weight loss (Fig 7B, left panel) and had lower average clinical scores, except for 9 days p.i., than ZH501-inoculated mice (Fig 7C, top panel). In 100 PFU group, mice inoculated with ZH501-P78-High virus showed delayed body weight loss (Fig 7B, right panel), as well as lesser and delayed onset of clinical symptoms than Zh501-inoculated mice (Fig 7C, bottom panel). Like MP-12, higher levels of P78 expression attenuated virulence of ZH501. Our data, thus, showed that P78 was a viral factor that attenuates RVFV virulence in mice.

## Discussion

By altering the sequence immediately upstream of the 1st AUG of the M segment RNA, we successfully rescued MP-12-derived and ZH-501-derived mutant viruses that expressed high levels of P78 in infected mammalian cells and carried high amounts of P78 in the virion. A past study using ZH501 demonstrated that P78 accumulation was higher in infected mosquito-derived C6/36 cells than in Vero E6 cells and that virions that were released from C6/36 cells, but not from Vero E6 cells, carried P78 [17]. The present and previous studies revealed that the intracellular accumulation levels of P78 correlated with the amounts of P78 in RVFV particles. Unlike the previous report [17], our recombinant ZH501 propagated in Vero E6 cells carried low levels of P78 in its virion (Fig 2). These different results may be due to the differences in virus preparation conditions and/or the sensitivity of antibodies used for detection of P78 in Western blot analysis.

We tested the replication kinetics of MP-12, P78-High virus, and P78-KO virus in seven different mammalian cell lines and a mosquito cell line (Fig 3). P78-KO virus replicated less efficiently than MP-12 throughout the infection in Vero E6 cells. As P78-KO virus showed lower virus binding efficiencies than MP-12 in Vero E6 cells (Fig 5), it is possible that inefficient binding of P78-KO virus to Vero E6 cells led to less efficient replication and lower infectivity (Fig 4) as compared with MP-12 in Vero E6 cells. P78-High virus exhibited more efficient replication than MP-12 in MEF, NIH-3T3, and Huh7 cells, whereas P78-KO virus showed similar replication kinetics to MP-12 in those cells and higher titers than MP-12 early in infection in MRC-5 cells. In Huh7 cells, P78-High virus replicated more efficiently than MP-12 (Fig 3) and had lower infectivity than MP-12 (Fig 4), while both showed no statistical differences in virus binding and internalization (Fig 5). These data suggest that efficient P78 expression, which occurred at the post-entry step, contributed to efficient replication of P78-High virus in this cell line.

Consistent with the notion that P78 promotes virus dissemination in mosquito *in vitro* and *in vivo* [18], P78-KO virus replicated less efficiently (Fig 3) and exhibited lower infectivity

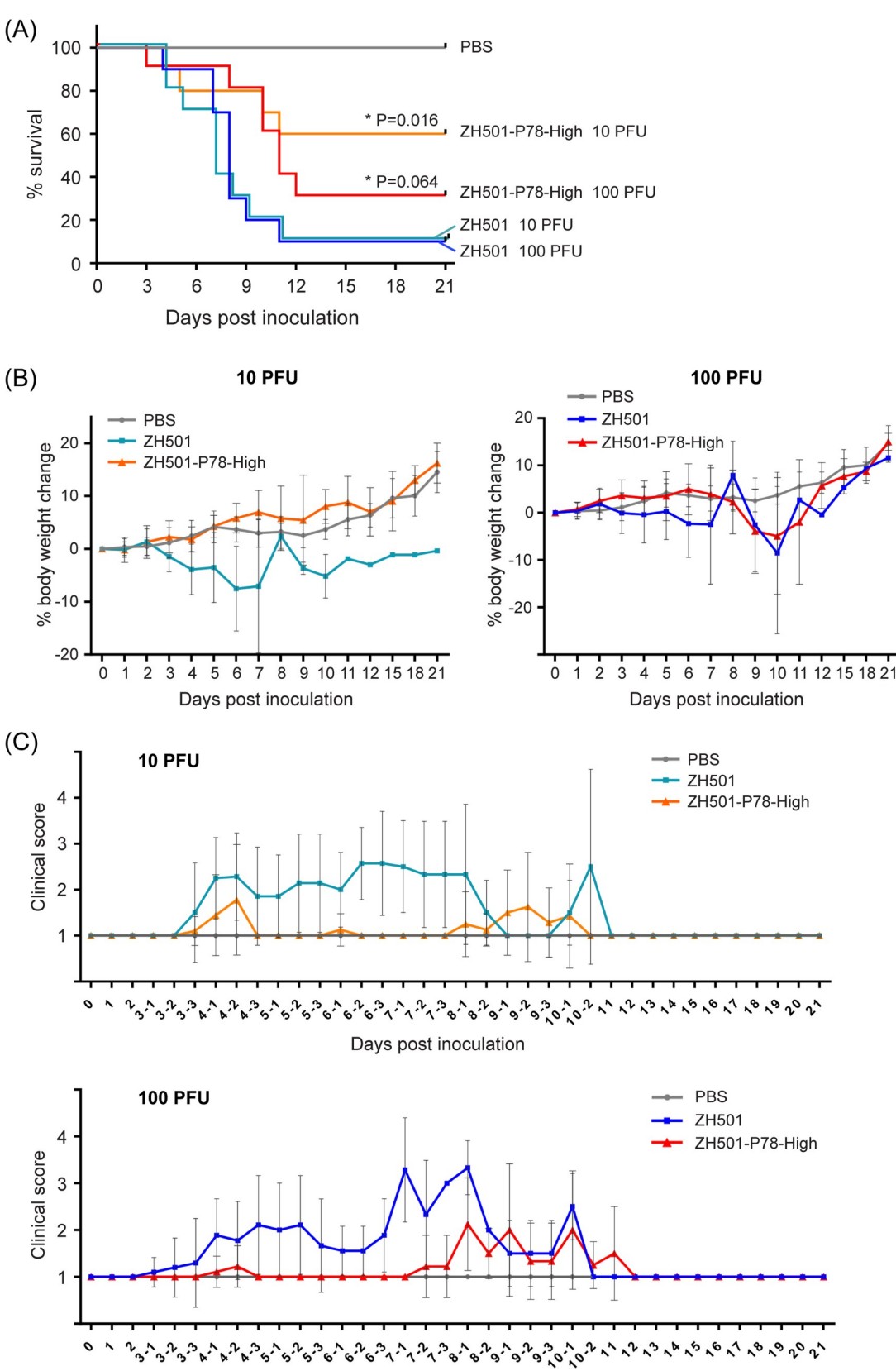

**Fig 7.** (A) Five-week-old CD-1 mice were inoculated with PBS or 10 PFU or 100 PFU of the indicated viruses intraperitoneally and observed for 21 days after inoculation. Differences in survival curves between the same dose of ZH501 and ZH501-P78-High were analyzed by log-rank test. (B) Body weight changes of mice inoculated with PBS or indicated viruses as a percentage compared to the day of inoculation. The data are mean ± SD. PBS data in 10 PFU and 100 PFU groups are the same data for comparison. (C) Clinical scores of the mice inoculated with PBS or indicated viruses. Animals were observed in the morning (shown in -1 after days p.i.), afternoon (shown in -2 after days p.i.), and evening (shown in -3 after days p.i.) from 3 to 10 days p.i. The data are mean ± SD. PBS data in 10 PFU and 100 PFU groups are the same data for comparison.

(Fig 4) than MP-12 in C6/36 cells. Unexpectedly, P78-High virus also replicated less efficiently and showed lower infectivity than MP-12 in C6/36 mosquito cells. Because the alteration of the sequence flanking the 1st AUG in P78-High virus was designed to express high levels of P78 in mammalian cells, the alteration of the nucleotide immediately upstream of 1$^{st}$ AUG might have negatively affected expression of P78 and/or Gn/Gc in mosquito cells, resulting in inefficient P78-High virus replication.

As compared with MP-12, P78-High virus and P78-KO virus replicated less efficiently and more efficiently, respectively, in Raw264.7 and PMA-treated THP-1 cells (Fig 3). Similarly, ZH501-P78-High virus replicated less efficiently than its parental ZH501 in Raw264.7 and PMA-treated THP-1 cells (Fig 3). These results demonstrated that the presence of P78 interfered with efficient RVFV replication in macrophage-derived cells. In addition to P78, M segment encodes major envelope proteins Gn/Gc, which are translated from the 4$^{th}$/5$^{th}$ AUGs of M mRNA. It appears that an increase in usage of the 1st AUG for P78 translation modestly inhibited translation efficiencies from the downstream AUGs, as Gn accumulation was slightly lower in P78-High virus-infected Vero E6 cells than in MP-12-infected cells (Fig 1C). Although lower Gn protein accumulation might have negatively affected virus replication in Raw 264.7 cells and PMA-treated THP-1 cells, P78-High virus replicated as efficiently as MP-12 or even higher than MP-12 in all other mammalian cell lines, suggesting that the effect of lower Gn expression had only a minor impact on virus replication. Virus binding assays showed that P78-High virus bound to Raw 264.7 cells less efficiently than MP-12, which probably led to inefficient internalization of P78-High virus in this cell line (Fig 5). The infectivity of P78-High virus was also lower than MP-12 in Raw 264.7 cells. These results suggested that P78-High virus failed to replicate as efficiently as MP-12 in Raw 264.7 cells because of its lower binding efficiency to the cells. However, virus binding efficiency was not the sole determinant for the differences in growth kinetics among the three viruses in Raw 264.7 cells, as P78-KO virus replicated better than MP-12 in this cell line (Fig 3), yet both viruses had a similar binding efficiency to Raw 264.7 cells (Fig 5).

RVFV infects human dendritic cells via dendritic cell-specific intercellular adhesion molecule-3-grabbing non-integrin (DC-SIGN) [30], a C-type lectin, which interacts with glycans and is expressed on dendritic cells and macrophages. THP-1 cells express human DC-SIGN, and Raw 264.7 cells express human DC-SIGN homologues, including SIGN-related 3, which is considered to have a similar function to human DC-SIGN [31,32]. Although it is unclear why P78-High virus bound to Raw 264.7 cells less efficiently than MP-12, it is possible that the presence of large amounts of P78, which encodes the extra preGn region compared with Gn, on the virion surface structurally affects the interaction between DC-SIGN and RVFV glycoproteins. Past structural analyses of RVFV particles and proteins primarily focused on Gn and Gc, but not P78 [10,12,33–35]. Further studies to solve the structure of RVFV particles carrying P78 would provide a clue as to the role of P78 at early stages of RVFV infection in both the insect and mammalian host.

P78-High virus exhibited significantly higher internalization efficiency compared with MP-12 in Vero E6 cells, even though attachment efficiencies were similar between the two viruses

(Fig 5). These results suggest that separate host factors are needed for virus attachment and virus internalization in cells lacking DC-SIGN and that the presence of more P78 on the virion surface facilitated virus internalization in some cells. Heparan sulfate serves as an attachment receptor for several viruses [36–38], while these viruses utilize different receptors for their internalization [39–41]. Likewise, heparan sulfate has been identified as a host factor for efficient RVFV infection, promoting virus binding to cells [42], while host factor(s) that plays a central role in internalization of RVFV have not yet been identified.

We tested whether P78 expression level affects virus virulence in mice (Figs 6 and 7). Taking advantage of the fact that MP-12 is moderately virulent in young mice [43], we examined the importance of P78 for MP-12 virulence using a young mouse model. The P78-High virus and P78-KO virus was less virulent and more virulent than MP-12, respectively. Likewise, ZH501 was more virulent than ZH501-P78-High virus in five-week-old CD1 mice (Fig 7). These studies demonstrated that P78 was a viral attenuation factor in mice. A previous study showed that the absence of P78 expression in a ZH548-derived mutant had little impact on virus virulence in C57BL/6 mice [18]. As the parental virus is already highly virulent, a possible increase in the virulence of a ZH548-derived mutant lacking P78 expression might have been masked in these studies. Alternatively, due to low P78 expression in wt RVFV (Fig 2), further reduction of P78 expression in wt RVFV might have a negligible impact on viral virulence. We noted a correlation between the replication efficiencies of MP-12, ZH501, and their mutants in macrophage cell lines (Fig 3) and their virulence in mice (Figs 6 and 7); viruses that replicated efficiently in these macrophage cells had higher virulence. These results support the possibility that differences in the replication levels of RVFV and its mutants in macrophages of infected mice contribute to differences in their virulence. Others have also suggested the contribution of macrophages to the pathogenesis of RVFV by demonstrating that depletion of macrophages resulted in prolonged survival time of Ifnar1-deficient mice challenged with a RVFV ZH548 mutant lacking NSs [44]. Also, several studies reported that RVFV infected macrophages *in vitro* and *in vivo* [45,46] and inhibited efficient production of proinflammatory cytokines [47]. Taken together, the present and past studies indicate an important link between efficient RVFV replication in macrophages and outcome of the disease.

## Acknowledgments

We thank Haitao Hu in the Department of Microbiology and Immunology, The University of Texas Medical Branch, for providing THP-1 cells.

## Author Contributions

**Conceptualization:** Kaori Terasaki.

**Formal analysis:** Kaori Terasaki, Birte Kalveram, Kendra N. Johnson, Terry Juelich, Jennifer K. Smith, Lihong Zhang.

**Funding acquisition:** Kaori Terasaki, Shinji Makino.

**Investigation:** Kaori Terasaki, Birte Kalveram, Kendra N. Johnson, Terry Juelich, Jennifer K. Smith, Lihong Zhang.

**Methodology:** Kaori Terasaki, Birte Kalveram, Kendra N. Johnson, Terry Juelich, Jennifer K. Smith, Lihong Zhang, Shinji Makino.

**Supervision:** Alexander N. Freiberg, Shinji Makino.

**Writing – original draft:** Kaori Terasaki, Shinji Makino.

**Writing – review & editing:** Kaori Terasaki, Birte Kalveram, Kendra N. Johnson, Terry Juelich, Jennifer K. Smith, Lihong Zhang, Alexander N. Freiberg, Shinji Makino.

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
