## [Decision Letter · Decision Letter 0]

23 Mar 2021

Dear Dr. Terasaki,

Thank you very much for submitting your manuscript "Rift Valley fever phlebovirus 78kDa envelope protein attenuates virus replication in macrophage-derived cell lines and viral virulence in mice" for consideration at PLOS Neglected Tropical Diseases. As with all papers reviewed by the journal, your manuscript was reviewed by members of the editorial board and by several independent reviewers. In light of the reviews (below this email), we would like to invite the resubmission of a significantly-revised version that takes into account the reviewers' comments. 

We cannot make any decision about publication until we have seen the revised manuscript and your response to the reviewers' comments. Your revised manuscript is also likely to be sent to reviewers for further evaluation.

Sincerely,

Jonas Klingström

Associate Editor

Esther Schnettler

Deputy Editor

Reviewer's Responses to Questions

**Key Review Criteria Required for Acceptance?**

**Methods**

-Are the objectives of the study clearly articulated with a clear testable hypothesis stated?

-Is the study design appropriate to address the stated objectives?

-Is the population clearly described and appropriate for the hypothesis being tested?

-Is the sample size sufficient to ensure adequate power to address the hypothesis being tested?

-Were correct statistical analysis used to support conclusions?

-Are there concerns about ethical or regulatory requirements being met?

Reviewer #1: Methods are clearly described and acceptable.

Reviewer #2: (No Response)

**Results**

-Does the analysis presented match the analysis plan?

-Are the results clearly and completely presented?

-Are the figures (Tables, Images) of sufficient quality for clarity?

Reviewer #1: See comments below.

Reviewer #2: (No Response)

**Conclusions**

-Are the conclusions supported by the data presented?

-Are the limitations of analysis clearly described?

-Do the authors discuss how these data can be helpful to advance our understanding of the topic under study?

-Is public health relevance addressed?

Reviewer #1: See comments below.

Reviewer #2: (No Response)

**Editorial and Data Presentation Modifications?**

Reviewer #1: (No Response)

Reviewer #2: (No Response)

**Summary and General Comments**

Reviewer #1: Overall the results presented are of high quality and make a significant contribution to the field. However, much of the work was performed with MP-12 even though the authors have produced P78-High virus in a ZH501 background. Thus, additional experiments using ZH501 P78-High (and P78-KO if possible) would strengthen the research. 

Major comments

1) The authors should examine the replication of P78-High in mosquito cells.

2) The authors should check replication kinetics of ZH501 P78-High (and P78-KO if possible) in at least Raw and THP-1 cells. This experiment is important as it will determine if increased expression of P78 also suppresses viral replication in macrophages with a virulent strain of RVFV. 

3) Given the conflicting results with ZH548 (line 523-525), it is important for the authors to test their ZH501 mutant viruses in mice. However, use of 16-day old mice may not be appropriate. Rather testing in older immune competent mice with lower doses of virus would enable an evaluation of P78-High and P78KO virulence. 

Minor comments:

1) As the authors have already published on P78-KO virus (reference 15) it would be helpful if they could indicate the previous name of the virus one time in the manuscript. 

2) Fig 1D: The ratio of p78 to Gn signal was discussed (lines 308-309) but is not shown in the figure. Please add to the figure.

Reviewer #2: In this manuscript, the authors suggest a novel insight into cell tropism of Rift Valley fever virus, which may be regulated by the expression of P78 viral glycoprotein and its incorporation into the virions. This study also reemphasizes that efficient replication of RVFV in macrophages may be the key determinant of virulence in vivo. Although these findings are of interest and potentially provide insights into novel functions of GP78 in RVFV replication and pathogenesis, the study is fairly descriptive; therefore, the following critical issues should be addressed to improve this manuscript prior to publication.

1. Figure 3, to examine the impact of high expression of GP78 on RVFV growth in mosquito cells, this reviewer strongly recommends the authors to perform a growth kinetics study in C6/36 cells. 

2. Figures 4 and 5, the authors should perform the experiments with more appropriate cell lines such as THP-1, Huh7 (for Fig. 5), and C6/36 to discuss the role(s) of GP78 in pathogenesis and cell/host tropism. 

3. Figure 6, to define the role of GP78 in macrophage tropism and virus growth ability in vivo (blood, liver, and spleen), the authors should perform (i) immunohistochemistry to detect RVFV antigens in macrophages and other parenchymal cells and compare the numbers of RVFV-positive macrophages among wild-type, GP78-high, and GP78-KO mutants.

PLOS authors have the option to publish the peer review history of their article (what does this mean?). If published, this will include your full peer review and any attached files.

Reviewer #1: No

Reviewer #2: No
---

## [Decision Letter · Decision Letter 1]

2 Sep 2021

Dear Dr. Terasaki,

We are pleased to inform you that your manuscript 'Rift Valley fever virus 78kDa envelope protein attenuates virus replication in macrophage-derived cell lines and viral virulence in mice' has been provisionally accepted for publication in PLOS Neglected Tropical Diseases.

Best regards,

Jonas Klingström

Associate Editor

Esther Schnettler

Deputy Editor

Reviewer's Responses to Questions

**Key Review Criteria Required for Acceptance?**

**Methods**

-Are the objectives of the study clearly articulated with a clear testable hypothesis stated?

-Is the study design appropriate to address the stated objectives?

-Is the population clearly described and appropriate for the hypothesis being tested?

-Is the sample size sufficient to ensure adequate power to address the hypothesis being tested?

-Were correct statistical analysis used to support conclusions?

-Are there concerns about ethical or regulatory requirements being met?

Reviewer #1: (No Response)

**Results**

-Does the analysis presented match the analysis plan?

-Are the results clearly and completely presented?

-Are the figures (Tables, Images) of sufficient quality for clarity?

Reviewer #1: (No Response)

**Conclusions**

-Are the conclusions supported by the data presented?

-Are the limitations of analysis clearly described?

-Do the authors discuss how these data can be helpful to advance our understanding of the topic under study?

-Is public health relevance addressed?

Reviewer #1: (No Response)

**Editorial and Data Presentation Modifications?**

Reviewer #1: (No Response)

**Summary and General Comments**

Reviewer #1: The authors have carefully considered the critiques and have performed additional experiments to strengthen the manuscript.

PLOS authors have the option to publish the peer review history of their article (what does this mean?). If published, this will include your full peer review and any attached files.

Reviewer #1: No

---

## [Editor Report · Acceptance letter]

8 Sep 2021

Dear Dr. Terasaki,

We are delighted to inform you that your manuscript, "Rift Valley fever virus 78kDa envelope protein attenuates virus replication in macrophage-derived cell lines and viral virulence in mice," has been formally accepted for publication in PLOS Neglected Tropical Diseases.

Best regards,

Shaden Kamhawi

co-Editor-in-Chief

Paul Brindley

co-Editor-in-Chief
